# Detection and evaluation of signals associated with exposure to individual and combination of medications in pregnancy: a signal detection study protocol

Anuradhaa Subramanian [1], Siang Ing Lee [1],
Sudasing Pathirannehelage Buddhika Hemali Sudasinghe,[1] Steven Wambua,[1]
Katherine Phillips [1], Megha Singh [1], Amaya Azcoaga-Lorenzo [2,3]
Neil Cockburn,[1] Jingya Wang [1], Adeniyi Fagbamigbe,[2,4] Muhammad Usman,[2]
Christine Damase-Michel,[5,6] Christopher Yau,[7] Lisa Kent [8], Colin McCowan,[2]
Dermot OReilly,[8] Gillian Santorelli,[9] Holly Hope,[10] Jonathan Kennedy [11],
Mohamed Mhereeg [12], Kathryn Mary Abel,[10,13] Kelly-Ann Eastwood,[8,14]
Mairead Black,[15] Maria Loane [16], Ngawai Moss,[17] Sinead Brophy [11],
Peter Brocklehurst,[1] Helen Dolk,[16] Catherine Nelson-Piercy,[18]
Krishnarajah Nirantharakumar [1], MuM-PreDiCT Group

PB, HD, CN-P and KN are joint senior authors.

For numbered affiliations see end of article.

**Correspondence to**
Krishnarajah Nirantharakumar;
k.nirantharan@bham.ac.uk

## ABSTRACT

**Introduction** Considering the high prevalence of polypharmacy in pregnant women and the knowledge gap in the risk–benefit safety profile of their often-complex treatment plan, more research is needed to optimise prescribing. In this study, we aim to detect adverse and protective effect signals of exposure to individual and pairwise combinations of medications during pregnancy.

**Methods and analysis** Using a range of real-world data sources from the UK, we aim to conduct a pharmacovigilance study to assess the safety of medications prescribed during the preconception period (3 months prior to conception) and first trimester of pregnancy. Women aged between 15 and 49 years with a record of pregnancy within the Clinical Practice Research Datalink (CPRD) Pregnancy Register, the Welsh Secure Anonymised Information Linkage (SAIL), the Scottish Morbidity Record (SMR) data sets and the Northern Ireland Maternity System (NIMATS) will be included. A series of case control studies will be conducted to estimate measures of disproportionality, detecting signals of association between a range of pregnancy outcomes and exposure to individual and combinations of medications. A multidisciplinary expert team will be invited to a signal detection workshop. By employing a structured framework, signals will be transparently assessed by each member of the team using a questionnaire appraising the signals on aspects of temporality, selection, time and measurement-related biases and confounding by underlying disease or comedications. Through group discussion, the expert team will reach consensus on each of the medication exposure–outcome signal, thereby excluding spurious signals, leaving signals suggestive of causal associations for further evaluation.

## STRENGTHS AND LIMITATIONS OF THIS STUDY

⇒ A comprehensive range of medication exposures prescribed both individually and as a combination with another medication within primary care is included in this study, providing a wide opportunity to detect and evaluate signals of adverse and protective effect during pregnancy.

⇒ This study utilises a wide range of data sources obtaining prescription data from primary care (from Clinical Practice Research Datalink and Secure Anonymised Information Linkage), and the community (from PIS), but is limited by the unavailability of prescription data from secondary care.

⇒ The results of this signal detection study are limited by the quality of routinely collected exposure and outcome data used for this study.

⇒ This signal detection study is susceptible to type 1 and type 2 errors owing to limitations in terms of multiple testing and insufficient sample size, respectively.

⇒ While the results of this exploratory signal detection study may provide useful signals, they may suffer from biases related to confounding and must be followed up by methodologically rigorous pharmacoepidemiological studies focused on each signal separately.

**Ethics and dissemination** Ethical approval has been obtained from the Independent Scientific Advisory Committee, SAIL Information Governance Review Panel, University of St. Andrews Teaching and Research Ethics Committee and Office for Research Ethics Committees

Northern Ireland (ORECNI) for access and use of CPRD, SAIL, SMR and NIMATS data, respectively.

## INTRODUCTION

Conventional methods of drug development from discovery to preclinical and clinical research and review are expensive, time consuming and disproportionately concentrate on disease areas that predominantly affect men.[1] The effect of this can be seen in the female dominance of adverse drug effect reporting, especially during the period of reproductive age.[2] Pregnant and breastfeeding women are further inclined to become 'therapeutic orphans', by their routine exclusion from clinical trials, given the struggle to insure such trials involving pregnant women. Pregnant women are further disadvantaged by having to arduously navigate through multifarious social influences, some losing their autonomy in medical decision making around taking medications or in their choice to participate in a clinical trial during pregnancy.[3 4]

In light of the urgent action required to develop safe medications for use during pregnancy, the 'Healthy Mum, Healthy Baby, Healthy Future' report by the Birmingham Health Partners advocates inclusion of pregnant women in clinical trials unless there are specific safety concerns.[5] The report also makes a case to pilot a 'Maternal Investigation Plan' similar to the 'Paediatric Investigation Plan' implemented by European Union, to make licensing compulsory for use of medications during pregnancy.

In the absence of safety signals for medication exposure during pregnancy from clinical trial data, information on teratogenicity is limited and is only prospectively collected from national surveillance data, available from resources such as UK Teratology Information Service (UKTIS).[6] This limited availability of robust evidence on medication use during pregnancy has led to women often being prescribed medications 'off-label' in practice.[7] Without appropriate medication safety information, the decision to continue, discontinue or switch their medication falls to the women themselves and their healthcare providers.[8]

We have previously found that the prevalence of polypharmacy in the UK, during the first trimester of pregnancy alone, has increased from 8.7% to 18.7% over the last two decades.[9] While polypharmacy itself may be essential and beneficial in the management of multiple chronic conditions in combination with pregnancy-related complications, inappropriate polypharmacy may cause preventable adverse drug events due to drug–drug interactions.[10] Considering the high prevalence of polypharmacy among women of reproductive age and the knowledge gap in safety profiles of complex treatment plan,[9 11–13] more research is urgently needed to optimise prescribing by detecting signals of adverse outcomes following exposure to combination of medications.

Real-world data (RWD) can be used to assess the post-market safety of the use of individual and combinations of medication during pregnancy, generating evidence to support regulatory decision-making. In addition, RWD also provides an opportunity for repurposing of approved medications as prophylactic treatments for pregnancy-related complications.[14] RWD-based signal detection involves a systematic step-by-step approach:[15–17] (1) identifying RWD source and inspecting its feasibility in accurately ascertaining exposures and outcomes to support identification of safety signals, (2) listing the outcomes (adverse events) and exposures (medications) of interest, (3) generating measures of statistical association for large sets of exposure–outcome pairs with methodological design to limit confounding, (4) reviewing identified signals by a multidisciplinary team and considering sources of bias that lead to false positive signals to ensure contextual interpretation and (5) strengthening and confirming screened signals using rigorous pharmaco-epidemiological studies or clinical trials.

The feasibility of mining RWD to detect adverse and protective effect signals of individual medications during pregnancy has already been established.[16 18] In this study, as part of the 'Multimorbidity and Pregnancy: Determinants, Clusters, Consequences and Trajectories (MuM-PreDiCT)' consortium, we aim to develop and implement a systematic signal detection methodology to determine adverse and protective signals from both individual and combinatorial medication exposure on the incidence of various pregnancy outcomes, using a set of large primary care, secondary care and maternity care databases within the UK.

## AIMS

The study aims to evaluate the feasibility of conducting a signal detection study estimating and evaluating the adverse and protective signals of medications prescribed during the preconception period and the first trimester of pregnancy on the incidence of various pregnancy outcomes using RWD.

The key objectives include the following:
1. To scope RWD sources in UK, and to check their feasibility for signal detection by exploring the range of medications prescribed and recorded within said data sources during the preconception period and first trimester of pregnancy.
2. To identify a core list of pregnancy outcomes, and to explore the feasibility of capturing these outcomes within said data sources
3. To apply suitably developed methods from published literature to estimate measures of disproportionality for each of the exposure–outcome pairs and detect signals.
4. To systematically review the detected signals through a multidisciplinary expert committee workshop, using a prespecified structured questionnaire exploring sources of false positives.

## METHODS AND ANALYSIS
### Data source
Four population-based data sources spanning all four nations of the UK will provide data for this signal detection

**Table 1** Scoping real-world data source from England to check its' feasibility for signal detection

| Data | Features of the data set | |
|---|---|---|
| CPRD | Setting | Primary care practices using the Vision software system |
| | Geographical region | All four nations of the UK, with relatively few practices heavily concentrated in three conurbations and the South |
| | Coverage | 4% of the UK |
| | Linkage availability to | ► Pregnancy Register and Mother–Baby Linked data.<br>► Deprivation data (patient postcode linked deprivation measures, practice postcode linked deprivation measures, Index of Multiple Deprivation (IMD), Townsend Deprivation Index, Carstairs Index, Rural–Urban Classification).<br>► Office of National Statistics (ONS) Death Registration Data.<br>► Hospital Episode Statistics (HES).<br>► Cancer Registration Data (however, this is not accessed for this study).<br>► Congenital anomaly register (however, this is not accessed for this study). |
| | Data content (with linkage) | ► Patient demographics.<br>► Consultation details.<br>► Symptoms, signs and diagnoses.<br>► Referrals to external care.<br>► Immunisation details.<br>► Records of test data in the GP system.<br>► All prescriptions (medications and appliances) issued by the GP. |
| | Period of data availability | January 2000 to July 2022 |
| | Feasibility to identify pregnancy episodes? | Yes, using CPRD Pregnancy Register |
| | Sample size—number of pregnancy episodes | 1.5 million pregnancies |
| | Definition of pregnancy start date | As derived within the CPRD Pregnancy Register |
| | Feasibility to identify exposures? | Yes, using prescription records on the GP system. This is coded using the Dictionary of Medicines and Devices (DM+D). Each product within the dictionary is associated with a BNF code to which the product belongs. The range of exposures for this study will be defined using the BNF code. Limitation: the data relate to primary care prescribing only. Secondary care prescribing data, over the counter medications and data on adherence to treatments are unavailable using this data source. Furthermore, absence of dispensing data may cause exposure misclassification in the instance of prescriptions that are issued but not dispensed. |
| | Feasibility to identify pregnancy outcomes? | Yes, using Read code diagnoses within primary care or ICD-10 diagnoses within linked HES data. |

BNF, British National Formulary; CPRD, Clinical Practice Research Datalink; GP, general practitioner; HES, Hospital Episode Statistics.

study. These include primary care data sources such as Clinical Practice Research Datalink (CPRD, from all four nations) and The Secure Anonymised Information Linkage (SAIL, Wales) and secondary care data sources such as Hospital Episode Statistics (HES, England), Scottish Morbidity Records (SMR) with linked community prescription data (Scotland) and Northern Ireland Maternity System (NIMATS) with linked Enhanced Prescribing Database (EPD, Northern Ireland). The data sources are scoped, and their feasibility for this signal detection study is assessed and tabulated in tables 1–4.

### Clinical Practice Research Datalink and Hospital Episode Statistics

CPRD Gold and Aurum contains anonymised, longitudinal medical records of over 20 and 39 million patients in the UK collected by over 985 and 1489 participating general practices, respectively, as part of their care and support.[19] It currently covers general practices that use the Vision and EMIS software, and collects data from

20% of general practices in the UK.[20] It includes data on demographics, diagnoses and prescriptions. Linkage to area-based deprivation index known as the Index of Multiple Deprivation (IMD) and Hospital Episodes Statistics (HES) is available for 75% of patients in England, whose general practices have consented to the CPRD linkage scheme.

CPRD Gold contains data on medications prescribed within primary care with associated prescription time stamps, encoded using drug codes that are assigned and categorised according to British National Formulary (BNF) item codes. However, CPRD Gold may be limited by their unavailability of secondary care and over the counter medication data, and data on whether these prescriptions were dispensed.

Within CPRD, the CPRD Pregnancy Register is an algorithm that takes information from maternity, antenatal and birth health records from primary care to detect

 

**Table 2** Scoping real-world data source from Wales to check its' feasibility for signal detection

| Data | Features of the data set | |
|---|---|---|
| SAIL | Setting | A population level database in Wales. SAIL is a repository of anonymised health and socioeconomic administrative data that provide linkage at an individual level using Trusted Third Party (TTP) in the NHS Wales Informatics Service (NWIS). |
| | Geographical region | Wales |
| | Coverage | The database contains 90% of all the GP data in Wales and 100% of all hospital admissions (PEDW) |
| | Linkage availability to | ▶ Primary care data (from Wales Longitudinal General Practice (WLGP)). <br> ▶ Secondary care data (from inpatient hospital admissions, inpatient from Patient Episode Database for Wales (PEDW) and outpatient from Outpatient Database for Wales (OPDW). <br> ▶ The National Community Child Health (NCCH). <br> ▶ The Maternal Indicators (MIDS). <br> ▶ The Welsh Dispensing Data Set (WDDS). <br> ▶ The Welsh Demographic Service Data set (WDSD). <br> ▶ Office of National Statistics (ONS) Annual District Birth Extract. <br> ▶ Office of National Statistics (ONS) Annual District Death Extract. <br> ▶ Congenital Anomaly Register and Information Service (CARIS). |
| | Data content (with linkage) | ▶ (WLGP): signs, symptoms, test results, diagnoses, prescribed treatment, referrals for specialist treatment. <br> ▶ (PEDW): attendance and clinical information, diagnoses and operations performed. (OPDW): attendance information for all NHS Wales hospital outpatient appointments. <br> ▶ (NCCHD): comprises information pertaining to birth registration, monitoring of child health examinations and immunisations. <br> ▶ (MIDS): contains data relating to the woman at initial assessment and to the mother and baby (or babies) for all births. <br> ▶ (WDDS): containing information on GP prescribed medications, dispensed medications by community contractors have been linked to the SAIL databank. <br> ▶ (WDSD): used to extract Lower-layer Super Output Area (LSOA) version 2011 information associated with area-level deprivation (the Welsh Index for Multiple Deprivation (WIMD) version 2019). <br> ▶ All births in Wales collected from birth registrations. <br> ▶ All deaths relating to Welsh residents, including those that died out of Wales. <br> Information about fetuses or babies who has or is suspected of having a congenital anomaly, |
| | Period of data availability | 2000–2022 |
| | Feasibility to identify pregnancy episodes? | Yes, using GP, PEDW, NCCHD and MIDS data sets |
| | Sample size—number of pregnancy episodes | ~475 000 pregnancies (from the year 2000 onwards) |
| | Definition of pregnancy start date | Pregnant women can be identified as any woman who have pregnancy codes in the WLGP data, in hospital admissions (PEDW) for pregnancy, or mothers in the NCCH or MIDS data with the baby birth date (pregnancy end date) and gestational age at birth available. |
| | Feasibility to identify exposures? | Yes, data cover prescriptions that are prescribed in Wales by GPs (general medical practitioners) and non-medical prescribers who have prescribed on behalf of the GP practice, that are then dispensed in the community within Wales. The data include all prescribed medicines, dressings and appliances that are dispensed each month. Information includes the number of prescription items dispensed by each community pharmacy in Wales, broken down by the GP practice in which they are prescribed and also the number of prescription items prescribed in each GP practice in Wales broken down by the pharmacy that dispensed those items. <br> Limitation: the data relate to primary care prescribing only. Secondary care prescribing data, over the counter medications and data on adherence to treatments are unavailable using this data source |
| | Feasibility to identify pregnancy outcomes? | Yes, using Read code diagnoses within primary care or ICD-10 diagnoses within hospital admissions data |

GP, general practitioner; NHS, National Health Service; SAIL, Secure Anonymised Information Linkage.

pregnancies and their outcomes.[21] A subset of the Pregnancy Register, Mother–Baby Linked (MBL) data further allows for studying outcomes in the children.

**The secure anonymised information linkage**

The SAIL databank, a population level database in Wales, is a repository of anonymised health and socioeconomic administrative data that provide linkage at an individual level.[22] It holds health data for 4.8 million people and includes data contributed by 80% of Welsh general practices. National Community Child Health Data set, GP records and hospital records have been used to detect pregnancies in SAIL.[23] In addition, patient level linkage to the Welsh Longitudinal General Practice data set and the Welsh Demographic Service data set has been used to obtain data on diagnoses, prescriptions and demographics data, respectively. The Welsh Dispensing Data

**Table 3** Scoping real-world data source from Scotland to check its' feasibility for signal detection

| Data | Features of the data set | |
|------|--------------------------|---|
| SMR | Setting | Scottish Morbidity Records (SMR) from secondary care data |
| | Geographical region | Scotland |
| | Coverage | All obstetric inpatients and day cases from maternity hospitals in Scotland |
| | Linkage availability to | ▶ Community prescriptions |
| | Data content (with linkage) | ▶ SMR00—outpatients<br>▶ SMR01—inpatient and day cases<br>▶ SMR02—maternity<br>▶ SMR04—mental health<br>▶ SMR06—cancer<br>▶ Accident and emergency<br>▶ National Records of Scotland (NRS) deaths<br>▶ Prescribing Information System (community)<br>▶ Scottish Birth Record (SBR)<br>▶ NRS infant deaths<br>▶ NRS stillbirths<br>▶ Scottish Linked Congenital Condition Data set (SLiCCD) (however, this is not accessed for this study) |
| | Period of data availability | SMR02 (pregnancy identification) from 2008 to 2021 with historical data from women<br>SMR01, SMR00, PIS and A&E database for the first 5 years of life on the children |
| | Feasibility to identify pregnancy episodes? | Yes |
| | Sample size—number of pregnancy episodes | 670, 811 births+pregnancies not ending in births |
| | Definition of pregnancy start date | Based on gestational age at booking appointment and or outcome of pregnancy |
| | Feasibility to identify exposures? | Yes, using prescription from the community. This is coded using the Dictionary of Medicines and Devices (DM+D). Each product within the dictionary is associated with a BNF code to which the product belongs. The range of exposures for this study will be defined using the BNF code.<br>Limitation: the data relate to primary care prescribing only. Secondary care prescribing data, over the counter medications and data on adherence to treatments are unavailable using this data source |
| | Feasibility to identify pregnancy outcomes? | Only secondary care |

PIS, Prescribing Information System.

Set (WDDS) containing information on general practitioner (GP) prescribed medications and dispensed medications by community contractors has been linked to the SAIL databank. Similar to CPRD, SAIL may be limited by its unavailability of secondary care and over the counter medication data and data on whether the prescriptions were dispensed.

### Scottish Maternity Records and linked data sources

The Scottish Maternity Records (SMR02) will be linked to data from Hospital Admissions (SMR01), Mental Health Inpatients (SMR04), Accident and Emergency and the Demography and Death registries,[23] covering diagnoses and demographic data for all inpatient stays and day cases for residents in Scotland. Maternity records (SMR02) or pregnancy-related hospital admissions (SMR01) allows for identification of pregnancies, and prescription data of medications dispensed in the community can be obtained from linked Prescribing Information System (PIS).

### Northern Ireland Maternity System

NIMATs holds demographic and clinical information on mothers and infants. It captures data relating to the current complete maternity process, but also contains details about the mother's medical and obstetric history. NIMATs contains information on medications that mothers self-reported to have taken during pregnancy. In addition to self-reported data on medications, linkage of NIMATs to EPD allows for analysis of prescription data issued by GPs.[24]

### Study design

A series of case control studies will be conducted to estimate measures of disproportionality, detecting signals of association between each of the pregnancy outcomes of interest and exposure to individual or combination of medications prescribed during the preconception period and first trimester of pregnancy.

### Study population

For this feasibility study, we aim to conduct the analyses separately across all four databases described in the section above. Women aged between 15 and 49 years with a pregnancy recorded within the CPRD Gold Pregnancy Register (all four nations), National Community Child Health data (Wales), SMR02 (Scotland) and NIMATS

**Table 4** Scoping real-world data source from Northern Ireland to check its' feasibility for signal detection

| Data | Features of the data set | |
|---|---|---|
| NIMATS | Setting | NIMATs is available in all five Health Trust areas across Northern Ireland (NI), within each hospital providing maternity services (11 hospitals in total). Access to NIMATS is also available to midwives/clerical staff in various community clinics across NI to allow for booking appointments to be recorded. |
| | Geographical region | Northern Ireland |
| | Coverage | 100% within Northern Ireland; full coverage from 2011 |
| | Linkage availability to | ▶ NHAIS (Health Register)<br>– GP Patient Registration Index<br>– Household characteristics (eg, single person households)<br>– Linkage to GRO—(maternal deaths; infant deaths (may be under-represented due to inability to link some infants to health records via Health and Care Number));<br>– Via postcode<br>– NI Multiple Deprivation Measure (NIMDM)—area level deprivation, usually supplied as quintile or decile.<br>– Settlement Band Classification for urban/rural classification (2005–2014) (postcode used only by HBS staff to aid linkage; not available to researchers).<br>– Land and Property Services—rateable value of property (another measure of deprivation).<br>▶ Enhanced Prescribing Database (EPD)—database includes information on all medications that have been prescribed by General Practitioners and dispensed to patients in NI.<br>▶ Secondary care data—includes hospital episodes from The Patient Administration System (PAS) and ED attendance data.<br>▶ NI Cerebral Palsy Register. |
| | Data content (with linkage) | ▶ Demographic information.<br>▶ Clinical information on mothers, pregnancies and infants—delivery, investigations, medications, hospital stays, postnatal complications.<br>▶ Mother's medical and obstetric history. |
| | Period of data availability | 2011–2022 |
| | Feasibility to identify pregnancy episodes? | Yes, the mother is identified by her unique Health and Care Number, but she has a different hospital maternity number for each pregnancy. |
| | Sample size—number of pregnancy episodes | ~300 000 pregnancies |
| | Definition of pregnancy start date | Can be derived from NIMATS variable EDC based on ultrasound scan |
| | Feasibility to identify exposures? | Linkage to EPD; NIMATS also records medications taken at booking interview (maternal self-report), however this is a free-text variable and prone to variations in spelling. It is also deemed potentially disclosive and therefore not released to the researcher. On request, HBS may be able to provide a 'dummy' variable for medications of particular interest.<br>Limitation: the data relate to primary care prescribing only. Secondary care prescribing data, over the counter medications and data on adherence to treatments are unavailable using this data source. |
| | Feasibility to identify pregnancy outcomes? | Recorded in NIMATS |

EDC, Estimated Date of Conception; GRO, General Register Office; HBS, Honest Broker Service; NIMATS, Northern Ireland Maternity System.

(Northern Ireland), within the study period customised to the period of data availability within each data source will be eligible for inclusion in this study (tables 1–4). Data standard quality checks for each of the databases, and eligibility criteria for inclusion is presented in a previous publication.[23] Pregnancy start dates will be either derived using a predefined algorithm or used as reported within the said data source (tables 1–4), and will be used to define exposure and outcome time windows.

### Outcomes

The MuM-PreDiCT and ConcePTION consortium have developed core outcome sets (a minimum set of recommended maternal and offspring outcomes) for studies of pregnant women with multiple long-term conditions and for studies generating medication safety evidence,

respectively.[25 26] The outcome set was reviewed by a study advisory group panel comprising of GPs, obstetricians, obstetric physicians and experienced users of the available data sources, to identify the pregnancy outcomes' suitability and feasibility for inclusion in this signal detection study. The availability, prevalence, quality and completeness of data recording of these outcomes within the said data sources were used as criteria to determine the feasibility of the outcome of interest. Outcomes with poor recording in a specific database may not be included in the analysis using that database to avoid noisy signals due to insufficient power or case misclassification. Furthermore, outcomes that cannot be considered as a medication adverse event such as termination of pregnancy, outcomes that were too broad or non-specific such

as involvement of patients in care decisions, and neonatal outcomes that were too narrow and reflective of prematurity such as intubation/ventilation requirement were excluded. The final list of outcomes to be considered for inclusion in this study is available in table 5, and the operational definitions of these outcomes is available in online supplemental table 1.

## Exposure

All medications prescribed within primary care (as in the CPRD Gold database (all four nations) and EPD (Northern Ireland)) and all medications dispensed in the community (as in the PIS database (Scotland)) have a BNF code associated with them in the UK. Dispensed medications in WDDS (Wales) are coded based on the Dictionary of Medicines and Devices (DM+D). Using a complete extract of the dictionary,[27] each dispensed item within the dictionary can be mapped to a corresponding BNF code.

Analysis stratified by BNF codes has been established in a previous analysis, where using a dictionary of medications prescribed within primary care, we stratified the medications prescribed according to their 4-digit BNF code (BNF chapter, section, paragraph and subparagraph), screened and selected 577 BNF items that were pharmacological agents with therapeutic action.[9]

Using a similar strategy, we will ascertain the exposure information for a range of medications stratified by their BNF code specifically within four crucial time windows: (1) preconception period (up to 90 days prior to the start of pregnancy), (2) first trimester of pregnancy (first 12 weeks of pregnancy), (3) second trimester of pregnancy (between 13 and 26 weeks of pregnancy) and (4) third trimester of pregnancy (between 27 weeks and end of pregnancy).[28] However, the exposure ascertainment within these windows will be restricted further to the time prior to outcome diagnosis to preserve exposure–outcome temporality. For outcomes that occur during the first trimester of pregnancy such as miscarriage, the exposure time window will be restricted to the preconception period and first trimester only. Furthermore, we will ascertain the exposure information for a range of medication pairs that are prescribed concurrently within the same exposure window to assess adverse and protective effect signals associated with medications prescribed in pairs.

## Covariates

The following demographic and health characteristics will be obtained from the four data sets; maternal age at the start of pregnancy, ethnicity, smoking status as recorded prior to the start of pregnancy, pregravid body mass index (BMI) and a wide range of comorbidities. Patients with missing data on smoking status, pregravid BMI and ethnicity will be categorised into a separate missing category within the corresponding variable. The list of 79 pre-existing long-term comorbidities for which baseline data were extracted, and their definitions is presented in a previous publication.[23] Ethnicity will be categorised as White, South Asian, Black Afro-Caribbean, mixed ethnic background and other ethnic minority groups including Chinese. Latest BMI recorded prior to the start of pregnancy will be considered as pregravid BMI, and will be categorised according to the WHO definition as underweight ($<18.5 \, kg/m^2$), healthy weight ($18.5–24.9 \, kg/m^2$), overweight ($25–29.9 \, kg/m^2$) and obese ($\geq 30 \, kg/m^2$).[29] Smoking status of patients prior to the start of pregnancy will categorise patients as current smokers, ex-smokers and never-smokers.

## Statistical analysis

### Description of baseline characteristics

Patient covariates will be summarised and stratified by their outcome status using numbers and percentages for categorical variables and mean (SD) or median (IQR) for continuous variables.

### Statistical analysis

OR will be considered as the primary measure of disproportionality, and will be estimated for each of the exposure–outcome pairs using a series of univariate logistic regression models. In a series of adjusted logistic regression models, the exposure–outcome relationships will be adjusted for covariates including age at start of pregnancy, pregravid BMI, ethnicity, smoking status and a disease risk score (DRS) to obtain adjusted ORs with 95% CIs. DRS will be generated for each of the outcomes using logistic regression models considering the outcome as a dependent variable and a range of pre-existing long-term conditions as independent variables. This is done to limit the effect of confounding attributable to prescriptions issued in order to manage the underlying long-term conditions. Relative decrease in p-value (p-RD) prior to and after adjustment for covariates will be calculated using the formula below.[16]

$$p - RD = \frac{p \, (before \, adj \, for \, DRS) - p \, (after \, adj \, for \, DRS)}{p \, (before \, adjustment)}$$

In addition to statistical measures of disproportionality, descriptive data stratified by their outcome status will be presented. These include numbers and proportions of eligible pregnancies with a prescription of the individual medication or medication combinations (pairs) during the two separate exposure windows.

Benjamini–Hochberg correction for multiple testing will be applied with a threshold of 0.20.

The analyses in each of the four data sets will be conducted separately. Signals arising from each of the four data sources will be reviewed side-by-side through a systematic signal review workshop described below.

The methods in this protocol are reported in line with RECORD (REporting of studies Conducted using Observational Routinely-collected health Data) guidelines (online supplemental table 2).

### Systematic signal review

A multidisciplinary expert team comprising of epidemiologists, GPs, obstetricians, obstetric physicians, pharmacists,

**Table 5** List of pregnancy outcomes included in this signal detection study from the core outcome set published by the ConcePTION and MuM-PreDiCT consortium

| List of core outcomes from ConcePTION consortium | Source of outcome documentation | | | |
|---|---|---|---|---|
| | CPRD | SAIL | NIMATS | SMR |
| Miscarriage | Data available as variable in CPRD Pregnancy Register | Data unavailable Currently, within SAIL researchers are unable to account for terminations as these are classed as sensitive data and not currently accessible for research purposes | Variable recorded in NIMATS, but only if it has occurred after booking appointment (early miscarriages may be under-represented) | Data available in SMR02 when women are admitted to secondary care |
| Intrauterine death/ stillbirth/perinatal death | Data available as variable in CPRD Pregnancy Register and linked HES data | Data available as stillbirths flagged in NCCH and Annual District Birth Extract (ABDE) data sets | IUD=yes Stillbirth=yes Perinatal death=partial, some infant deaths will not be captured in NIMATS but may be picked up in General Register Office (GRO) or (Patient Administration System) PAS data | Data available in SMR02 |
| Small for gestational age (SGA) | Birth weight, baby's sex and gestational age data available within CPRD Pregnancy Register and linked HES data, from which SGA can be derived | Can be derived from birth weight data available in ADBE and NCCH, gestational age available in the latter | Yes—variable 'Birth Centile' available 2016 onwards If missing, can be derived from NIMATS variables (birth weight, sex and gestational age at delivery) | Can be derived from variables recorded in SMR02 (birth weight, sex and gestational age at delivery) |
| Preterm birth | Gestational age available within CPRD Pregnancy Register from which preterm birth can be derived | Data available in NCCH | Can be derived from variables recorded in NIMATS (gestational age at delivery, EDC) | Can be derived from variables recorded in SMR02 (gestational age at delivery, EDC) |
| Overall congenital anomalies (CA) including termination of pregnancy due to fetal anomaly | Data available in primary care data, and linkage to baby records identified from mother–baby linked (MBL) data | Data available from the Congenital Anomaly Register and Information Service (CARIS) database and the NCCH_SIG_COND data set which is used to inform it. | Yes—NIMATS (variable=infant complications; value='congenital abnormality') | Data unavailable |
| Specific major congenital anomalies (not including termination of pregnancy due to fetal anomaly) | Congenital anomalies recorded with sufficient quality will be included. Data available in primary care data, and linkage to baby records identified from MBL data | Data available in CARIS database | Data unavailable. Data in hospital discharge records are not suitable for identifying specific congenital anomalies. | Data unavailable |
| Maternal death | Data available in primary care data | Data available from GP/ Patient Episode Data for Wales (PEDW)/ADDE for death during childbirth | Data available in GRO | Data available in National Records of Scotland |
| Additional outcomes from MuM-PreDiCT consortium | | | | |
| Maternal outcomes | | | | |
| Clinical: antenatal | | | | |
| Pre-eclampsia, eclampsia, HELLP syndrome, gestational hypertension | Data available in primary care and linked HES data | Data available in primary care and linked PEDW data (if diagnosed) | Data available in the delivery details description, present pregnancy problems and induction reason variables in NIMATS, as well as diagnoses variable in PAS | Recorded from chapter XV of ICD10 Pregnancy, Childbirth and the Puerperium (O00-O99). |

Continued

**Table 5** Continued

| List of core outcomes from ConcePTION consortium | Source of outcome documentation | | | |
|---|---|---|---|---|
| | CPRD | SAIL | NIMATS | SMR |
| Placenta abruption | Data available in primary care and linked HES data | Data available in primary care and linked PEDW data (if diagnosed) | Data available in the delivery details description, indication for caesarean section variables in NIMATS, as well as diagnoses variable in PAS | Recorded from chapter XV of ICD10 Pregnancy, Childbirth and the Puerperium (O00–O99) |
| Venous thromboembolism | Data available in primary care and linked HES data | Data available in primary care and linked PEDW data (if diagnosed) | Data available as postnatal complications variable in NIMATS and as a diagnoses variable in PAS | Recorded from chapter XV of ICD10 Pregnancy, Childbirth and the Puerperium (O00–O99). |
| **Clinical: peripartum** | | | | |
| Preterm premature rupture of membrane (PPROM) | Data available in linked HES data | Data available in primary care and linked PEDW data (if recorded) | Data available as postnatal complications variable in NIMATS and as a diagnoses variable in PAS | Recorded from chapter XV of ICD10 Pregnancy, Childbirth and the Puerperium (O00–O99). |
| Severe maternal morbidity | Specific maternal morbidities recorded with sufficient quality will be included. Data available from primary care data | Data available in primary care and linked PEDW data (if recorded) | Data available as specific maternal morbidities in NIMATS | Recorded from chapter XV of ICD10 Pregnancy, Childbirth and the Puerperium (O00–O99). |
| Postpartum haemorrhage | Data available in linked HES data | Data available in primary care and linked PEDW data (if recorded) | Data available as postnatal complications variable in NIMATS and as a diagnoses variable in PAS | Recorded from chapter XV of ICD10 Pregnancy, Childbirth and the Puerperium (O00–O99). |
| **Mental health** | | | | |
| Self-harm/suicide | Data available in primary care and linked HES data | Data available in primary care, linked PEDW and emergency admissions data | Data available in Self-harm and Suicide Ideation Register | Data available in National Records of Scotland |
| Postpartum mental illness | Data available in primary care and linked HES data | Data available in primary care and linked PEDW data (if diagnosed) | Data available as postnatal complications variable in NIMATS and as a diagnoses variable in PAS | Recorded from chapter XV of ICD10 Pregnancy, Childbirth and the Puerperium (O00–O99). |
| **Children's outcomes** | | | | |
| Cerebral palsy/ autism/ADHD/ neurodevelopmental outcomes | Data available in primary care data, and linkage to baby records identified from MBL data | Data available in primary and linked PEDW data | Partial data available in NI Cerebral Palsy Register (NICPR) | Data unavailable |

ADHD, Attention Deficit Hyperactivity Disorder; CPRD, Clinical Practice Research Datalink; EDC, Estimated Date of Conception; HELLP, Haemolysis, Elevated Liver enzymes and Low Platelet; HES, Hospital Episode Statistics; IUD, Intrauterine Device; MuM-PreDiCT, Multimorbidity and Pregnancy: Determinants, Clusters, Consequences and Trajectories; NCCH, National Community Child Health; NIMATS, Northern Ireland Maternity System; SAIL, Secure Anonymised Information Linkage; SMR, Scottish Morbidity Record.

data scientists, experts in genetics, internal medicine and pharmacovigilance and researchers with expertise on the outcome of interest will be invited to a signal detection workshop. The workshop is aimed at collaboratively working and screening potential signals to identify and exclude signals that are likely to be affected by bias and confounding, leaving signals suggestive of causal associations to be further evaluated in future studies.

A list of all the exposure–outcome pairs, the adjusted and unadjusted OR along with 95% CIs and p-RD will be presented to the multidisciplinary team in the order of statistical significance and strength of association. Exposure–outcome pairs with a clinically significant strength of association without statistical significance will also be provided to the review team to avoid false negatives.

A checklist with the following items will be provided to the review team: (1) confirmation of exposure–outcome temporality to limit the possibility of reverse causality,[30] (2) consideration of concomitant medications to limit the possibility of coprescription bias,[31] (3) consideration medical history, (4) consideration of demographic features and (5) consideration of underlying disease and other alternative explanations to limit the possibility of confounding. After consideration of the checklist items, we will request each reviewer to mark the exposure–outcome pairs with three possible response options: (1) 'already established', (2) 'warrant further investigation' and (3) 'dismissed', with reasons for each response. In a group discussion, conflicting responses will be discussed, and a consensus will be made.

## Patient and public involvement

Patient and public involvement (PPI) has been extensive from design to dissemination of research outputs from the MuM-PreDiCT group. Our PPI representatives have provided advice on the importance and relevance of this study and helped shape the research question. PPI representatives were also involved in screening and identifying the pregnancy outcomes of interest that are relevant to this study. One of the PPI representatives (NM) also coauthored this protocol. PPI representatives will be involved in the interpretation of the results in the future.

## Ethics approval and consent to participate

CPRD has ethics approval from the Health Research Authority to support research using anonymised patient data. Use of CPRD and linked HES data for this study is approved by the Independent Scientific Advisory Committee. Use of SAIL databank for this study is approved by the SAIL Information Governance Review Panel. Use of SMR data for this study is approved by the School of Medicine Ethics Committee, acting on behalf of the University of St. Andrews Teaching and Research Ethics Committee. SAIL and SMR data will be analysed within a Safe Haven Research environment. Use of NIMATS and EPS data for this study is approved by the Office for Research Ethics Committees Northern Ireland (ORECNI) and the Honest Broker Governance Board.

## Consent for publication

As the study data will be deidentified, consent for publication is not required.

**Author affiliations**
[1]Institute of Applied Health Research, University of Birmingham, Birmingham, UK
[2]Division of Population and Behavioural Sciences, University of Saint Andrews School of Medicine, St. Andrews, Fife, UK
[3]Hospital Rey Juan Carlos. Research Network on Chronicity, Primary Care and Health Promotion-RICAPPS (RICORS), Instituto de Investigación Sanitaria Fundación Jimenez Diaz, Madrid, Spain
[4]Department of Epidemiology and Medical Statistics, University of Ibadan, Ibadan, Nigeria
[5]Medical and Clinical Pharmacology, School of Medicine, Université Toulouse III, Toulouse, France
[6]Center for Epidemiology and Research in Population Health (CERPOP), INSERM, Toulouse, France
[7]Nuffield Department of Women's and Reproductive Health, University of Oxford, Oxford, UK
[8]Centre for Public Health, Queen's University Belfast, Belfast, UK
[9]Born In Bradford, Bradford Institute for Health Research, Bradford, UK
[10]Centre for Women's Mental Health, Faculty of Biology Medicine & Health, University of Manchester, Manchester, UK
[11]Data Science, Medical School, Swansea University, Swansea, UK
[12]Swansea University Medical School, Swansea, UK
[13]Manchester Mental Health & Social Care Trust, Manchester, UK
[14]University Hospitals Bristol and Weston NHS Foundation Trust, Bristol, UK
[15]Aberdeen Centre for Women's Health Research, School of Medicine, Medical Science and Nutrition, University of Aberdeen, Aberdeen, UK
[16]The Institute of Nursing and Health Research, University of Ulster, Belfast, UK
[17]Queen Mary University of London, London, UK
[18]Guy's & St Thomas' Foundation Trust, London, UK

**Collaborators** MuM-PreDiCT Group: Rachel Plachcinski, Shakila Thangaratinam, Beck Taylor, Astha Anand, Richard Riley, Jonathan Ian Kennedy, Mohamed Mhereeg, Louise Locock, Zoe Vowles, Neil Cockburn, Francesca Crowe, Sharon Mccann, Charles Gadd, Stephanie Hanley, Luciana Rocha Pedro

**Contributors** Our authors list includes PPI coinvestigator NM. AS, SIL, KP, AA-L, CD-M, CM, DOR, HH, JK, KMA, K-AE, MB, ML, NM, SB, PB, HD, CN-P and KN conceived the study and contributed to the study design. AS led the development of the protocol and drafted the initial manuscript with contribution from SIL, SPBHS, SW, KP, MS, JW, LK and JK, and supervision from AA-L, CY, CD-M, CM, DOR, KMA, MB, ML, SB, PB, HD, CN-P and KN. AA-L, NC, AF, MU, CD-M, LK, GS, JK and MM provided advise on the real-world data sources and their feasibility for use in this study. All authors critically reviewed and revised the protocol drafts and agreed on the final draft manuscript for submission.

**Funding** This work is independent research funded by the Strategic Priority Fund 'Tackling multimorbidity at scale' programme (grant number MR/W014432/1) delivered by the Medical Research Council and the National Institute for Health Research in partnership with the Economic and Social Research Council and in collaboration with the Engineering and Physical Sciences Research Council. The views expressed are those of the author and not necessarily those of the funders, the NIHR or the UK Department of Health and Social Care. The funders had no role in study design, data collection and analysis, decision to publish or preparation of the manuscript. This work was also supported by Health Data Research UK (HDRUK2023.0030), which is funded by UK Research and Innovation, the Medical Research Council, the British Heart Foundation, Cancer Research UK, the National Institute for Health and Care Research, the Economic and Social Research Council, the Engineering and Physical Sciences Research Council, Health and Care Research Wales, Health and Social Care Research and Development Division (Public Health Agency, Northern Ireland), Chief Scientist Office of the Scottish Government Health and Social Care Directorates

**Competing interests** AS has no declarations during the study period; after the study was completed, she has left the University of Birmingham and taken a post in AstraZeneca. The other authors declare no competing interests.

**Patient and public involvement** Patients and/or the public were involved in the design, or conduct, or reporting or dissemination plans of this research. Refer to the Methods section for further details.

**Patient consent for publication** Not required.

**Provenance and peer review** Not commissioned; externally peer reviewed.

**ORCID iDs**
Anuradhaa Subramanian http://orcid.org/0000-0001-8875-7363
Siang Ing Lee http://orcid.org/0000-0002-2332-5452
Katherine Phillips http://orcid.org/0000-0003-0674-605X
Megha Singh http://orcid.org/0000-0003-3680-7124
Amaya Azcoaga-Lorenzo http://orcid.org/0000-0003-3307-878X
Jingya Wang http://orcid.org/0000-0003-1498-2693
Lisa Kent http://orcid.org/0000-0002-8882-0526
Jonathan Kennedy http://orcid.org/0000-0002-1122-6502
Mohamed Mhereeg http://orcid.org/0000-0003-1241-9549
Maria Loane http://orcid.org/0000-0002-1206-3637
Sinead Brophy http://orcid.org/0000-0001-7417-2858

Krishnarajah Nirantharakumar http://orcid.org/0000-0002-6816-1279

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
