## [Reviewer comments · BMJ Open]

ARTICLE DETAILS

TITLE (PROVISIONAL)	Detection and evaluation of signals associated with exposure to individual and combination of medications in pregnancy: A signal detection study protocol
AUTHORS	Subramanian, Anuradhaa; Lee, Siang Ing; Hemali Sudasinghe, Sudasing Pathirannehelage Buddhika; Wambua, Steven; Phillips, Katherine; Singh, Megha; Azcoaga-Lorenzo, Amaya; Cockburn, Neil; Wang, Jingya; Fagbamigbe, Adeniyi; Usman, Muhammad; Damase-Michel, Christine; Yau, Christopher; Kent, Lisa; McCowan, Colin; O'Reilly, Dermot; Santorelli, Gillian; Hope, Holly; Kennedy, Jonathan; Mhereeg, Mohamed; Abel, Kathryn; Eastwood, Kelly-Ann; Black, Mairead; Loane, Maria; Moss, Ngawai; Brophy, Sinead; Brocklehurst, Peter; Dolk, Helen; Nelson-Piercy, Catherine; Nirantharakumar, Krishnarajah

VERSION 1 – REVIEW

REVIEWER	John Brock Harris Wingate University
REVIEW RETURNED	14-May-2023

GENERAL COMMENTS	Thank you for the opportunity to review the work. Overall: The methodology is sound and is presented in a streamlined manner. A few overarching considerations are below. Consider phrasing the work in the present tense instead of future tense. Example: Abstract line 44: approval will be obtained or is obtained? There are instances where a comma is used in a series of two items and in series of 3 or more items but not in all series of 3 or more. Examples: page 7 lines 11, 13, 27, and 42; page 8 lines 40, 42, and 44 (no all inclusive) Assure all acronyms are defined with first use. Examples: UK defined on page 6 line 43 but used throughout page 5 and GP on page 7 line 37, EPD line 10 page 8, and BNF on line 60 on page 8 (not all inclusive) another is WHO on page 9 line 40. Consider using medications instead of drugs throughout the work. In some instances drugs are considered illicit in the literature. Consider using combination of medications (line 18 page 8) versus combined medications (line 45 page 5). The intent seems to be multiple medication regimens versus a single agent containing 2+
---

	agents. Combination of medications throughout the work would help assure a reader understands the differences. Make sure the dates of the study are in the work. Specific section considerations are below. Introduction: Combined medications page 5 lines 45 and 49. Page 5 line 53: feasibility "in" accurately or feasibility "and" accurately? both read appropriately but have slightly different meanings. Page 5 line 59: and consider sources or and considering sources? Methods: page 6 line 43: UK defined here but used in previous pages. Page 7 (1), (2), and (3) headers are formatted differently. Page 7 line 10: that uses or that use? Page 7 line 27: pregnancy episodes. When reading, this phrase did not flow. Consider pregnancies. Page 7 lines 28-30: consider removing the statement. The statement after citation 21 is redundant to the sentence prior. Page 7 line 30: pregnancy register or Pregnancy Register? presented both ways in the paragraph. Page 7 line 36: data that provides or data that provide? Page 7 lines 39-40: With SAIL....to detect pregnancies, the flow is choppy when reading. Consider making this statement direct and active. GP and hospital records have been used to detect pregnancies using the National Community Child Health Dataset in SAIL. Page 7 lines 42 and 45: dataset have and contractors have to dataset has and contractors has. Page 7 line 46: their or its? Page 7 line 55: registry or registries? Page 8 line 8: history or histories? Study Population: Page 8 lines 29 and 33: database or databases? window or windows? Page 8 line 32: or used as "is" reported versus or used as reported? Outcomes: page 8 line 47: end statement after interest. Remove in this signal detection study. page 8 line 51: were too were too. Remove one were too.
--	--

	Page 8 line 52: prematurity such "a" versus prematurity such "as"? Exposure: Page 8 lines 57-60: consider removing the information in between the [and]. That information has been presented in previous sections. Page 8 and 9: BNF item code or BNF code or BNF Item Code? All 3 are used. please be consistent. Page 9 line 9: remove the phrase (recorded in the CPRD Gold). page 9 line 18: extra space between . and citation 28 Covariates: page 9 line 35 and lines 45-47: consider moving info from lines 45-47 up to the info in line 35. page 9 line 37: does Chinese not fall under Asian? Why is there a specific notation of Chinese? page 9 line 40 and 42: BMI of 25 is captured in both or not captured in both healthy and overweight. 24.9 is considered upper end of healthy by WHO. Page 9 line 41: Does the WHO BMI classification need a citation? Statistical Analysis: page 9 line 53: number or numbers? page 9 lines 55 and 56: standard deviation and interquartile range should be lower case. page 10 line 5: Odds ratio or Odds ratios? page 10 line 27: presented stratified? consider reworking this statement. page 10 line 46: OR not defined previously. page 10 line 49: remove high and low ORs. page 10 line 50: marginally statistically insignificance. p values are not measures of closeness to significance. Consider reworking this statement. A second look at those signals that are not significant is worth it. But not because it is "close" to significance. Page 10 lines 54 and 59: exposure-outcome or exposure - outcome? Ethics Page 11 lines 9-21: consider making this section active and direct using present tense. Again, thank you for the opportunity to review the work.
--	---

REVIEWER	Meredith Howley Birth Defects Research Section, New York State Department of Health
-----------------	--

REVIEW RETURNED	12-Jun-2023
-------------

GENERAL COMMENTS	This protocol outlines plans for a signal detection study using existing data sources in the UK. The authors outline the data sources, exposures, outcomes, and the calculation of ORs. They also provide a framework for how these ORs will be reviewed by a group of experts and move forward with investigation of signals in case-control studies. The focus will be specifically on prescription medications that are prescribed by GPs. I think the protocol is well-thought out and exciting. A few very small comments/considerations:  1. On page 9, Line 14/15: The authors talk about the two time periods of interest for exposures are the preconception period (90 days before pregnancy) and the first trimester. I certainly agree that these are important exposure windows for many of the outcomes listed in Table 5. I was curious their reasoning on the use of these windows as it relates to some of the outcomes in Table 5. For example, preterm birth and small-for-gestation-age might also very likely be impacted by medication use after the first 12 weeks of pregnancy. 2. Table 1-4: The row heading "feasibility to identify of exposures" might not need the word "of". Additionally, I noticed that Table 1 and 3 include limitations but Tables 2 and 4 do not. Some of the limitations listed in Tables 1 and 3 would apply to the other data sources. For example, none collect information on over-the-counter prescriptions. It could be helpful to be consistent in description the limitations across these tables. 3. Page 10, line 46/47: The authors talk about how the ORs will be reviewed by the multidisciplinary team. They describe that the ORs will be examined based on their relative decrease in p-value. I was curious if the authors had considered using the magnitude of the OR in any way. Potentially high signals may be less precise if the exposure/outcomes are rare, but might it be worth considering really high ORs (defined as over a certain OR value) that were high in both the crude and adjusted analyses that do not meet the traditional p-value difference cut off? 4. Minor comment: The abstract is missing a word in the second paragraph of the "Methods and analysis" section: "A series of case control studies will be conducted to estimate measures of disproportionality, detecting signals of association between a range of pregnancy [outcomes?], and exposure to individual and combinations of drugs."
--

VERSION 1 – AUTHOR RESPONSE

Reviewer 1

Overall: The methodology is sound and is presented in a streamlined manner.

A few overarching considerations are below.

Comment 1: Consider phrasing the work in the present tense instead of future tense. Example:

Abstract line 44: approval will be obtained or is obtained?

Response 1: We have now amended the "Ethics" section in the abstract and the "Ethics approval and consent to participate" section in the manuscript as per the reviewer's suggestion. These sections now reflect the current status of the work, with ethical approval in place to conduct the analyses.

Comment 2: There are instances where a comma is used in a series of two items and in series of 3 or more items but not in all series of 3 or more. Examples: page 7 lines 11, 13, 27, and 42; page 8 lines 40, 42, and 44 (no all inclusive)

Response 2: Thank you for this comment. We have now checked and amended throughout the manuscript where necessary. When the conjunction “and” joins two clauses, we have used comma before “and”. When the sentence contains a series of three or more words, phrases, or clauses, we have inserted a comma to separate the elements.

Comment 3: Assure all acronyms are defined with first use. Examples: UK defined on page 6 line 43 but used throughout page 5 and GP on page 7 line 37, EPD line 10 page 8, and BNF on line 60 on page 8 (not all inclusive) another is WHO on page 9 line 40.

Response 3: Thank you for this comment. We have now used the full form of the acronyms during their first use within the manuscript.

Comment 4: Consider using medications instead of drugs throughout the work. In some instances drugs are considered illicit in the literature.

Response 4: We agree with the reviewer. Throughout the manuscript, we have changed the word ‘drug’ to ‘medication’ where appropriate.

Comment 5: Consider using combination of medications (line 18 page 8) versus combined medications (line 45 page 5). The intent seems to be multiple medication regimens versus a single agent containing 2+ agents. Combination of medications throughout the work would help assure a reader understands the differences.

Response 5: Thank you for this comment. We have now used the phrase “combination of medications” instead of “combined medications” throughout the manuscript. However, please note that in addition to detecting and evaluating signals of individual medications, we intend to detect and evaluate signals for medications prescribed in pairs. The absence of the prescription of interest, or the prescription combination (pair) of interest within the exposure window will form the reference group in the two sets of analyses respectively. Please see a snippet from the exposure section below detailing the exposure to medications prescribed in pairs.

“Furthermore, we will ascertain the exposure information for a range of medication pairs that are prescribed concurrently within the same exposure window to assess adverse and protective effect signals associated with medications prescribed in pairs.”

Comment 6: Make sure the dates of the study are in the work.

Response 6: The study period is customised to the data availability in each dataset and is described in Table 1.

CPRD – 2000-2022

SAIL – 2000-2022

SMR – 2008-2021

NIMATS – 2011-2022

Comment 7: Specific section considerations are below.

Introduction:

- Combined medications page 5 lines 45 and 49.
- Page 5 line 53: feasibility "in" accurately or feasibility "and" accurately? both read appropriately but have slightly different meanings.
- Page 5 line 59: and consider sources or and considering sources?

Response 7: Thank you for these comments. The following revisions have been made to the manuscript based on suggestions.

- We have now changed “combined medications” to “combinations of medications” in page 5, lines 45-49
- In page 5, line 53, we mean that the real-world data sources will be scoped to check if the exposures and outcomes are recorded, i.e., their “feasibility in accurately ascertaining exposures and outcomes to support identification of safety signals”
- We have now revised the sentence to “...(4) reviewing identified signals by a multidisciplinary team and considering sources of bias that lead to false positive signals to ensure contextual interpretation...”

Comment 8:Methods:

- page 6 line 43: UK defined here but used in previous pages.
- Page 7 (1), (2), and (3) headers are formatted differently.
- Page 7 line 10: that uses or that use?
- Page 7 line 27: pregnancy episodes. When reading, this phrase did not flow. Consider pregnancies.
- Page 7 lines 28-30: consider removing the statement. The statement after citation 21 is redundant to the sentence prior.
- Page 7 line 30: pregnancy register or Pregnancy Register? presented both ways in the paragraph.
- Page 7 line 36: data that provides or data that provide?
- Page 7 lines 39-40: With SAIL....to detect pregnancies, the flow is choppy when reading. Consider making this statement direct and active. GP and hospital records have been used to detect pregnancies using the National Community Child Health Dataset in SAIL.
- Page 7 lines 42 and 45: dataset have and contractors have to dataset has and contractors has.
- Page 7 line 46: their or its?
- Page 7 line 55: registry or registries?
- Page 8 line 8: history or histories?

Response 8: Thank you for these comments. The following revisions have been made to the manuscript based on suggestions.

- UK is now defined where it appears first in the text now.
- The headers are now formatted correctly
- The sentence in page 7, line 10 now reads as “It currently covers general practices that use the Vision and EMIS software..”
- The sentence in page 7, line 27 now reads as “Within CPRD, the CPRD Pregnancy Register is an algorithm that takes information from maternity, antenatal and birth health records from primary care to detect pregnancies and their outcomes”
- The redundant sentence in page 7, line 29-30 is now removed.
- Throughout the manuscript, “Pregnancy Register” is now used as standard
- The sentence in page 7, line 36 now reads as “The SAIL databank, a population level database in Wales, is a repository of anonymized health and socio-economic administrative data that provide linkage at an individual level”
- The sentence in page 7, line 39-40 now reads as “National Community Child Health Dataset, GP records and hospital records have been used to detect pregnancies in SAIL”
- The sentences in page 7, line 42-45 now reads as “In addition, patient level linkage to the Welsh Longitudinal General Practice dataset and the Welsh Demographic Service dataset has been used to obtain data on diagnoses, prescriptions and demographics data respectively. The Welsh Dispensing Data Set (WDDS) containing information on general practitioner (GP) prescribed medications and dispensed medications by community contractors has been linked to the SAIL databank”

- The sentence in page 7, line 46 now reads as “Similar to CPRD, SAIL may be limited by its unavailability of secondary care and over-the-counter prescription data and data on whether the prescriptions were dispensed.”
- The sentence in page 7, line 55 now reads as “The Scottish Maternity Records (SMR02) will be linked to data from Hospital Admissions (SMR01), Mental Health Inpatients (SMR04), Accident and Emergency and the Demography and Death registries...”
- Since history is an uncountable noun, we have left the sentence as it is

Comment 9: Study Population:

- Page 8 lines 29 and 33: database or databases? window or windows?
- Page 8 line 32: or used as "is" reported versus or used as reported?

Response 9: Thank you for these comments. The following revisions have been made to the manuscript based on suggestions.

- The sentence in page 8, lines 29 now reads as “Data standard quality checks for each of the databases, and eligibility criteria for inclusion is presented in a previous publication”
- The sentence in page 8, line 32 now reads as “Pregnancy start dates will be either derived using a pre-defined algorithm or used as reported within the said data source (Table 1-4), and will be used to define exposure and outcome time windows.”

Comment 10: Outcomes:

- page 8 line 47: end statement after interest. Remove in this signal detection study.
- page 8 line 51: were too were too. Remove one were too.
- Page 8 line 52: prematurity such "a" versus prematurity such "as"?

Response 10: Thank you for these comments. The following revisions have been made to the manuscript based on suggestions.

- The sentence in line page 8, line 47 now reads as “The availability, prevalence, quality, and completeness of data recording of these outcomes within the said data sources were used as criteria to determine the feasibility of the outcome of interest”
- The duplicate ‘were too’ within this sentence has now been removed.
- In page 8, line 52, the phrase “prematernity such a intubation/ventilation requirement” has been replaced by “prematernity such as intubation/ventilation requirement” in this sentence.

Comment 11: Exposure:

- Page 8 lines 57-60: consider removing the information in between the [and]. That information has been presented in previous sections.
- Page 8 and 9: BNF item code or BNF code or BNF Item Code? All 3 are used. please be consistent.
- Page 9 line 9: remove the phrase (recorded in the CPRD Gold).
- page 9 line 18: extra space between . and citation 28

Response 11: Thank you for these comments. The following revisions have been made to the manuscript based on suggestions.

- We would like to keep the content in page 8, lines 57-60 as it is, since it provides some context for the exposure section
- Through the manuscript, “BNF Item code(s)” have been changed to “BNF code(s)”
- In page 9, line 9, the phrase “(recorded in CPRD Gold)” has now been removed
- All double spaces have now been replaced with single space within the manuscript.

Comment 12: Covariates:

- page 9 line 35 and lines 45-47: consider moving info from lines 45-47 up to the info in line 35.

- page 9 line 37: does Chinese not fall under Asian? Why is there a specific notation of Chinese?
- page 9 line 40 and 42: BMI of 25 is captured in both or not captured in both healthy and overweight. 24.9 is considered upper end of healthy by WHO.
- Page 9 line 41: Does the WHO BMI classification need a citation?

Response 12: Thank you for these comments. The following revisions have been made to the manuscript based on suggestions.

- As recommended by the reviewer, we have now moved the sentence in page 9, line 45-47 to page 9, line 35
- We have now corrected the ethnic categories and revised the sentence. The sentence in page 9, line 37 now reads as “Ethnicity will be categorized as White, South Asian, Black Afro-Caribbean, mixed ethnic background, and other ethnic minority groups including Chinese.”
- We have now corrected the categorisation of BMI and revised the sentence. The sentence in page 9, line 40-42 now reads as “Latest BMI recorded prior to the start of pregnancy will be considered as pre-gravid BMI, and will be categorized according to the World Health Organization definition as underweight (<18.5 kg/m²), healthy weight (18.5-24.9 kg/m²), overweight (25-29.9 kg/m²), and obese (≥30 kg/m²).”
- We have now added a reference for WHO classification of BMI

Comment 13: Statistical Analysis:

- page 9 line 53: number or numbers?
- page 9 lines 55 and 56: standard deviation and interquartile range should be lower case.
- page 10 line 5: Odds ratio or Odds ratios?
- page 10 line 27: presented stratified? consider reworking this statement.
- page 10 line 46: OR not defined previously.
- page 10 line 49: remove high and low ORs.
- page 10 line 50: marginally statistically insignificant. p values are not measures of closeness to significance. Consider reworking this statement. A second look at those signals that are not significant is worth it. But not because it is "close" to significance.
- Page 10 lines 54 and 59: exposure-outcome or exposure - outcome?

Response 13: Thank you for these comments. The following revisions have been made to the manuscript based on suggestions.

- The sentence in page 9, line 53 now reads as “Patient covariates will be summarized and stratified by their outcome status using numbers and percentages for categorical variables and mean (Standard Deviation) or median (Interquartile Range) for continuous variables.”
- In page 9, line 55-56, “Standard Deviation” and “Interquartile Range” have been changed to “standard deviation” and “interquartile range” respectively
- In page 10, line 5, the sentence refers to odds ratio as a measure of disproportionality
- In page 10, line 27, we have revised the sentence to “In addition to statistical measures of disproportionality, descriptive data stratified by their outcome status will be presented. These include descriptive measures of numbers and proportions of eligible pregnancies with a prescription of the individual medication or medication combinations (pairs) during the two separate exposure.”
- OR is now defined as Odds ratio during its first use within the manuscript
- In page 10, line 9, “(high and low ORs)” has now been removed from the sentence
- We agree with the reviewer. In page 10, line 50, the sentence is now revised and reads as “Exposure-outcome pairs with a clinically significant strength of association without statistical significance will also be provided to the review team to avoid false negatives.”
- In page 10, line 54-59, the hyphen joins the words exposure and outcome, suggesting pairing of the two, so we will retain “exposure-outcome pairs”

Comment 14: Ethics

- Page 11 lines 9-21: consider making this section active and direct using present tense.

Response 14: Thank you for this comment. We have now used present tense and active sentences in this section. The section now reads as “CPRD has ethics approval from the Health Research Authority to support research using anonymised patient data. Use of CPRD and linked HES data for this study is approved by the Independent Scientific Advisory Committee. Use of SAIL databank for this study is approved by the SAIL Information Governance Review Panel. Use of SMR data for this study is approved by the School of Medicine Ethics Committee, acting on behalf of the University of St. Andrews Teaching and Research Ethics Committee. SAIL and SMR data will be analysed within a Safe Haven Research environment. Use of NIMATS and EPS data for this study is approved by the Office for Research Ethics Committees Northern Ireland (ORECNI) and the Honest Broker Governance Board.”

Reviewer 2

This protocol outlines plans for a signal detection study using existing data sources in the UK. The authors outline the data sources, exposures, outcomes, and the calculation of ORs. They also provide a framework for how these ORs will be reviewed by a group of experts and move forward with investigation of signals in case-control studies. The focus will be specifically on prescription medications that are prescribed by GPs. I think the protocol is well-thought out and exciting. A few very small comments/considerations:

Comment 1: On page 9, Line 14/15: The authors talk about the two time periods of interest for exposures are the preconception period (90 days before pregnancy) and the first trimester. I certainly agree that these are important exposure windows for many of the outcomes listed in Table 5. I was curious their reasoning on the use of these windows as it relates to some of the outcomes in Table 5. For example, preterm birth and small-for-gestation-age might also very likely be impacted by medication use after the first 12 weeks of pregnancy.

Response 1: Thank you for these comments. We agree with the reviewer that many of medications prescribed during the second and third trimester could also impact the odds of some of the outcomes, therefore we have added the second and third trimester of pregnancy as exposure time windows to be included in the study. The exposure section is revised as follows:

“...we will ascertain the exposure information for a range of medications stratified by their BNF code specifically within four crucial time windows: (1) preconception period (up to 90 days prior to the start of pregnancy) (2) first trimester of pregnancy (first 12 weeks of pregnancy), (3) second trimester of pregnancy (between 13 and 26 weeks of pregnancy), and (4) third trimester of pregnancy (between 27 weeks and end of pregnancy). However, the exposure ascertainment within these windows will be restricted further to the time prior to outcome diagnosis to preserve exposure-outcome temporality. For outcomes that occur during the first trimester of pregnancy such as miscarriage, the exposure time window will be restricted to the preconception period and first trimester only..”

Comment 2: Table 1-4: The row heading “feasibility to identify of exposures” might not need the word “of”. Additionally, I noticed that Table 1 and 3 include limitations but Tables 2 and 4 do not. Some of the limitations listed in Tables 1 and 3 would apply to the other data sources. For example, none collect information on over-the-counter prescriptions. It could be helpful to be consistent in description the limitations across these tables.

Response 2: Thank you for this comment. We have removed the heading “feasibility to identify of exposures” and “feasibility to identify of pregnancy outcomes” to “feasibility to identify exposures” and “feasibility to identify pregnancy outcomes”. We have also added the limitations around identifying exposures to all the relevant tables (1-4) consistently.

Comment 3: Page 10, line 46/47: The authors talk about how the ORs will be reviewed by the multidisciplinary team. They describe that the ORs will be examined based on their relative decrease in p-value. I was curious if the authors had considered using the magnitude of the OR in any way. Potentially high signals may be less precise if the exposure/outcomes are rare, but might it be worth considering really high ORs (defined as over a certain OR value) that were high in both the crude and adjusted analyses that do not meet the traditional p-value difference cut off?

Response 3: Thank you for this comment. Kindly note that the signals presented to the reviewer will be based on the strength of association (i.e., the ORs) rather than the relative decrease in p-value. We have also now added a sentence that signals that are clinically significant but may not have reached statistical significance will also be presented to the signal review team. A snippet of this section below:

“A list of all the exposure-outcome pairs, the adjusted and unadjusted OR along with 95% confidence intervals, and p-RD will be presented to the multidisciplinary team in the order of statistical significance and strength of association. Exposure-outcome pairs with a clinically significant strength of association without statistical significance will also be provided to the review team to avoid false negatives.”

Comment 4: Minor comment: The abstract is missing a word in the second paragraph of the "Methods and analysis" section: "A series of case control studies will be conducted to estimate measures of disproportionality, detecting signals of association between a range of pregnancy [outcomes?], and exposure to individual and combinations of drugs."

Response 4: Thank you for pointing this out. We have now revised the sentence.

“A series of case control studies will be conducted to estimate measures of disproportionality, detecting signals of association between a range of pregnancy outcomes and exposure to individual and combinations of medications.”